# Targeting the Maladaptive Effects of Binge Drinking on Circadian Gene Expression

**DOI:** 10.3390/ijms231911084

**Published:** 2022-09-21

**Authors:** Kolter Grigsby, Courtney Ledford, Tanvi Batish, Snigdha Kanadibhotla, Delaney Smith, Evan Firsick, Alexander Tran, Kayla Townsley, Kaylee-Abril Vasquez Reyes, Katherine LeBlanc, Angela Ozburn

**Affiliations:** 1Portland Veterans Affairs Medical Center, Research and Development Service, Portland, OR 97239, USA; 2Department of Behavioral Neuroscience, Oregon Health and Science University, Portland, OR 97239, USA

**Keywords:** circadian, alcohol, nucleus accumbens, suprachiasmatic nucleus, ventral tegmental area, gene expression

## Abstract

Previous studies (1) support a role of circadian genes in regulating alcohol intake, and (2) reveal that harmful alcohol use alters circadian rhythms. However, there is minimal knowledge of the effects of chronic alcohol processes on rhythmic circadian gene expression across brain regions important for circadian biology and alcohol intake. Therefore, the present study sought to test the effects of chronic binge-like drinking on diurnal circadian gene expression patterns in the master circadian pacemaker (SCN), the ventral tegmental area (VTA), and the nucleus accumbens (NAc) in High Drinking in the Dark-1 (HDID-1) mice, a unique genetic risk model for drinking to intoxication. Consistent with earlier findings, we found that 8 weeks of binge-like drinking reduced the amplitude of several core circadian clock genes in the NAc and SCN, but not the VTA. To better inform the use of circadian-relevant pharmacotherapies in reducing harmful drinking and ameliorating alcohol’s effects on circadian gene expression, we tested whether the casein kinase-1 inhibitor, PF-67046, or the phosphodiesterase type-4 (an upstream regulator of circadian signalling) inhibitor, apremilast, would reduce binge-like intake and mitigate circadian gene suppression. PF-67046 did not reduce intake but did have circadian gene effects. In contrast, apremilast reduced drinking, but had no effect on circadian expression patterns.

## 1. Introduction

Alcohol Use Disorders (AUDs) have a devastating impact on individuals and their families and exert a high cost to society [1,2]. Binge drinking is a strong predictor of AUD diagnosis and has deleterious health consequences. Among the health deficits associated with harmful alcohol use are a persistent disruption of the sleep/wake cycle [3] and disruptions in rhythmic physiological processes, such as body temperature and melatonin release [4,5]. The molecular clock underlying circadian rhythms is an evolutionarily conserved process that allows organisms to adapt to their environment within a ~24 hr rhythm and is responsive to ethanol in humans and in rodents [6,7]. These rhythms are maintained by a series of transcriptional and translational feedback loops (TTFLs) [8]. The core circadian transcriptional regulators are *BMAL-1* (brain and muscle Arnt-like protein-1), *CLOCK* (circadian locomotor output cycles kaput), and *NPAS2* (neuronal PAS domain protein 2). *NPAS2* is structurally and functionally homologous to CLOCK and its role in the nucleus accumbens (NAc) is transcriptionally important for dopamine receptor regulation and drug reward [9]. The TTFL begins with BMAL-1 heterodimerization with either *CLOCK* or *NPAS2* to regulate the expression of PERIOD (*Per1*, *Per2*, and *Per3*), cryptochrome (*Cry1* and *Cry 2*), and thousands of other genes important for various biological functions, including neurotransmission, energy homeostasis, and immune signalling [10,11]. Following transcription and translation, *Per* and *Cry* proteins are post-translationally modified by casein kinase 1 (Ck1), and then translocate back to the nucleus and inhibit *CLOCK*/*NPAS2-BMAL1* mediated transcription, and thus, their own expression [8,12,13].

Two weeks of two-bottle choice ethanol drinking in mice, followed by one week of abstinence, resulted in a sustained reduction in expression of the circadian genes *Clock* and *Npas2* in the NAc and the ventral tegmental area (VTA), two brain regions well known for their importance in ethanol drinking and preference. In contrast, a recent study found that acute (4 days) binge-like ethanol intake increased *Clock* in the NAc shell sub-region and that knocking down NAc *Clock* reduced binge-like ethanol drinking in male C57BL/6J mice [14]. Circadian genes are not only responsive to ethanol but are thought to play a major role in alcohol drinking behaviors. Genetic variations in *Clock* and *Per2* have been associated with increased alcohol consumption in humans [7,12,15]. Moreover, mice with mutations in circadian genes (*Clock*, *Bmal1*, *Per1*, *or Per2)* exhibit increased ethanol intake, further supporting the importance of molecular clocks in regulating ethanol intake [12,16].

It is difficult to know whether the heightened ethanol drinking displayed by mutant mice results from circadian genes directly or from unknown compensatory mechanisms, such as downstream effects on reward-related processing. However, site-specific genetic manipulations provide additional support for understanding the role of circadian genes in alcohol consumption. VTA specific knockdown of *Clock* expression resulted in increased two-bottle choice ethanol drinking [12]. Moreover, selective ablation of *Bmal1* in striatal neurons increased two-bottle choice ethanol drinking in male mice and decreased intake in female mice, where the effect in males was shown to be *Per2*-dependent [16]. Notable differences in the drinking paradigms (two-bottle choice versus limited access, binge-like drinking), control groups (water versus sucrose access) and chronicity of ethanol intake suggests additional work is needed to better understand the relationship between the molecular clock and harmful alcohol use (and to produce more generalizable findings). To this end, we sought to test the importance of chronic binge-like ethanol drinking on circadian gene expression across a network of brain regions important for rhythms and reward, and to determine whether pharmacologically targeting circadian pathways would reduce binge-like ethanol drinking in a unique rodent genetic risk model for drinking to intoxication, the High Drinking in the Dark (HDID-1) mice.

Ethanol-induced neuroadaptations in circadian and other related molecular pathways are thought to be critical in AUD progression. Therefore, pharmacologically targeting circadian-related signaling to reduce harmful alcohol use may be a viable treatment option for AUDs. It is clear that *Per2* plays an important role in the period of molecular clock rhythms, a parameter of rhythms that is shortened with alcohol exposure [6]. One way to lengthen the period of the molecular clock is to inhibit the two primary isoforms of Casein kinase 1-epsilon and delta (*CK1e*/*d*; serine/threonine kinases important for circadian TTFL negative feedback) [17]. The *CK1e*/*d* inhibitor PF-670462 was shown to blunt the ‘Alcohol Deprivation Effect’ in male rats [18]. Another rhythmic parameter altered by alcohol is amplitude, where chronic alcohol decreased circadian gene expression (in the NAc and VTA, as described above) [12,19]. There is an established role of cAMP activity in gating circadian gene amplitude and rhythmicity in the suprachiasmatic nucleus (SCN), the master circadian regulator [20]. Moreover, cAMP-dependent signaling is considered a core component of the mammalian circadian pacemaker [21]. PDE4 is a known a regulator of cAMP and pharmacological inhibition of PDE4 inhibitor (rolipram) has been shown to increase rhythmic expression of *Per2* in cell culture [22]. PDE4 has recently gained research interest as a potential molecular target for treating AUD [23,24,25,26]. For instance, we recently demonstrated that PDE4 inhibition reduced binge-like drinking in HDID-1 mice when delivered peripherally (rolipram and apremilast) or site-specifically into the NAc [25,27]. Therefore, it follows that the capacity of PDE4 inhibition to reduce harmful drinking may have a basis in the rescue or amelioration ethanol-induced reductions in circadian gene amplitude.

HDID mice are a valuable genetic animal model with construct (maintains descriptive power for human AUD, i.e., accurately captures drinking to pharmacological intoxicating), face (resembles human binge drinking), and predictive (used to accurately predict human binge drinking outcomes) validity [27,28,29,30]. These unique mice offer a significant advantage for increasing the generalizability and translatability of animal research findings to humans. Using the HDID-1 mice and the DID task, the primary goals of the present study were (1) to determine whether chronic binge-like ethanol drinking alters circadian gene expression in the NAc, SCN, and VTA, and (2) test whether pharmacologically manipulating the molecular clock (via PF-670462 or apremilast) would reduce binge drinking and rescue ethanol-induced disruptions in circadian gene expression.

## 2. Results

### 2.1. Experiment 1: Chronic Binge-like Ethanol Drinking Reduces NAc and SCN Expression of Circadian Genes

To determine whether chronic binge-like ethanol drinking altered rhythmic levels of circadian gene expression across the NAc, VTA, or SCN, female HDID-1 mice were given limited access to 20% ethanol or water for 8-weeks. The average 4 h ethanol intake for weeks 1–8 was 4.43 ± 0.21 g/kg and the average 4 h water intake over these weeks was 60.22 ± 2.02 mL/kg. Descriptive statistics (including F and *p*-values) for each timepoint and brain region are provided in Table 1. For the **NAc**, a two-way ANOVA revealed a main effect of ethanol (fluid type) on the expression of *Clock* [F(1, 80) = 21.61; *p* < 0.0001], *Npas2* [F(1, 80) = 18.01; *p* < 0.0001], *Per1* [F(1, 80) = 34.67; *p* < 0.0001], *Per2* [F(1, 80) = 7.03; *p* < 0.01], and *Per3* [F(1, 80) = 5.28; *p* < 0.05], a main effect of ZT for all circadian genes [F-values are provided in Table 2; *p*’s < 0.05] and a fluid type x ZT interaction for *Clock*, *Bmal1*, and *Per2* (*p*’s < 0.05) (Figure 1A–F). A Bonferroni’s multiple comparison analysis revealed that binge-like ethanol intake reduced the NAc expression of *Clock* at ZT 9 and 18 (*p*’s < 0.01), and *Npas2* at ZT 15 and 18 (*p*’s < 0.05), *Per1* at ZT 6 and 18 (*p*’s < 0.05), and *Per2* at ZT 15 and 18 (*p*’s < 0.05). No significant effects were observed between fluid groups, ZT, nor were there any fluid type x ZT interactions in the **VTA**. For the **SCN**, a two-way ANOVA revealed a main effect of fluid type on the expression of *Per1* [F(1, 74) = 9.08; *p* < 0.01] and *Per2* [F(1, 76) = 12.22; *p* < 0.001], a main effect of ZT on the expression of *Bmal1* [F(7, 76) = 3.15; *p* < 0.01], *Per1* [F(7, 74) = 3.96; *p* < 0.001], *Per2* [F(7, 76) = 5.87; *p* < 0.0001], and *Per3* [F(7, 80) = 4.98; *p* < 0.001], and a fluid type x ZT interaction for the expression of *Per1*. Bonferroni’s multiple comparison testing revealed that chronic binge-like ethanol intake reduced the SCN expression of *Per1* at ZT 12 (*p* < 0.0001).

### 2.2. Experiment 2A: PF-670462 Did Not Reduce Chronic Binge-like Ethanol Intake

To determine the effects of *CK1e*/*d* inhibition on chronic binge-like ethanol drinking, 0 mg/kg or 30 mg/kg of PF-670462 was administered 30-min prior to a 4-day DID for 2 weeks following a 6-week baseline drinking period (experimental details are provided in Figure 2A). Analysis of the average 2 h ethanol intake during Days 1–3 for weeks 7 and 8 revealed no main effects of treatment or treatment x day interactions (Figure 2B). Similarly, analysis of the average 4 h ethanol intake on Day 4 of DID for weeks 7 and 8 showed no main effect of treatment (Figure 2B). A one-way, repeated measures ANOVA revealed a main effect of time [F(13, 1363) = 133.9; *p* < 0.001]. Tukey’s post hoc testing showed that the average ethanol intake across treatment groups was lower in weeks 7 and 8 compared to week 6. This suggests a handling effect, whereby i.p. injections alone reduced ethanol intake.

### 2.3. Experiment 2B: Inhibiting CK1E/D Reduces NAc Circadian Gene Expression

In experiment 1, we found that chronic harmful drinking reduced circadian gene expression in the NAc and SCN of HDID-1 mice. Moreover, circadian gene expression showed a biphasic “peak” and “trough” nature; therefore, expression analysis focused on ZT 3 (trough) and 15 (peak). To determine whether pharmacologically targeting circadian expression could rescue ethanol-induced gene disruption in the NAc, VTA and SCN, the *CK1e*/*d* inhibitor PF-670462 was given prior to a 4-day DID for 2-weeks (following 6-weeks of baseline drinking). For the **NAc**, a two-way ANOVA analysis revealed a main effect of treatment for *Per2* [F(1, 22) = 4.94; *p* < 0.05], *Per3* [F(1, 22) = 6.17; *p* < 0.05], and *Npas2* [F(1, 22) = 5.81; *p* < 0.05], a main effect of ZT for *Per1* [F(1, 22) = 64.84; *p* < 0.0001], *Per2* [F(1, 22) = 83.76; *p* < 0.0001], *Per3* [F(1, 22) = 42.02; *p* < 0.0001], *Clock* [F(1, 22) = 9.39; *p* < 0.01], and *Bmal1* [F(1, 22) = 23.83; *p* < 0.0001], and a treatment x ZT interaction for Per3 [F(1, 22) = 6.92; *p* < 0.05]. Tukey’s multiple comparison testing revealed that *Per2* expression was significantly reduced in ethanol-treated mice compared to the 0 mg/kg treated mice at ZT 3 (Figure 3E). When collapsing across ZT, PF-670462 treatment trended to increase NAc *Npas2* expression (*p* = 0.09; Figure 3B). For the **VTA**, there were no main effects of treatment, ZT, or treatment x ZT interactions. For the SCN, a two-way ANOVA analysis revealed a main effect of ZT for *Per1* [F(1, 22) = 24.80; *p* < 0.0001], *Per2* [F(1, 22) = 62.67; *p* < 0.0001], *Per3* [F(1, 22) = 70.08; *p* < 0.0001], *Clock* [F(1, 22) = 63.45; *p* < 0.0001], *Npas2* [F(1, 22) = 5.17; *p* < 0.05], and *Bmal1* [F(1, 22) = 11.25; *p* < 0.01], with no main effects of treatment or treatment x ZT interactions for any **SCN** gene tested (Figure 3 M–R).

### 2.4. Experiment 3A: Apremilast Reduces Ethanol Intake in iHDID-1 Mice with a History of Chronic Binge-like Drinking

Apremilast is a promising pharmacotherapeutic for treating AUDs and has been shown to reduce harmful drinking across established drinking paradigms [25]. Moreover, PDE4 inhibition has been shown to increase *Per2* expression in culture [22]. To determine whether apremilast would reduce excessive drinking in a circadian gene-driven manner, apremilast (0 or 40 mg/kg) was given prior to each drinking day in a 4-day DID for 2 weeks following a 6-week baseline period (identical to Experiment 2). Experimental details are provided in Figure 3A. The average 2 h ethanol intake during Days 1–3 during weeks 7 and 8 revealed a significant main effect of treatment [F(3, 88) = 27.20; *p* < 0.0001], and ZT [F(1, 82) = 4.33; *p* < 0.05] (Figure 4B). A two-way ANOVA showed no treatment x sex interactions. Therefore data were collapsed across sex. Tukey’s post hoc analysis revealed that apremilast treatment significantly reduced average 2 h ethanol for weeks 7 and 8 (*p*’s < 0.01) (Figure 4B). Analysis of the average 4 h intake on Day 4 during weeks 7 and 8 revealed a main effect of treatment [F(1, 82) = 53.61; *p* < 0.0001], no main effect of time, and a treatment x time interaction [F(1, 82) = 7.08; *p* < 0.0001]. There was no effect of sex or a treatment x sex interaction. Therefore data were collapsed on sex. Tukey’s multiple comparison testing revealed that apremilast treatment reduced 4 h ethanol intake during on day 4 of week 7 and for all drinking days in week 8 (*p*’s < 0.05) (Figure 4B). Similar to the observation in experiment 2, we saw a main effect of time across both apremilast treatment groups [F(31, 2760) = 83.99; *p* < 0.001]. Tukey’s post hoc testing found that that the average ethanol intake during weeks 7 and 8 was lower than week 6 for both apremilast treatment groups, again suggesting a handling effect in response to i.p. injections.

### 2.5. Experiment 3B: Apremilast Treatment Did Not Ameliorate the Effects of Ethanol on Circadian Gene Expression

To further evaluate the effects of pharmacologically targeting the molecular clock on rescuing ethanol-induced blunting of circadian gene expression, apremilast (0 or 40 mg/kg) was administered 30-min prior to DID. For the **NAc**, there were no main effects of treatment, nor treatment x ZT or treatment x sex interactions on circadian gene expression, therefore data were collapsed on sex. Analysis revealed a main effect of ZT for *Per2* [F(1, 73) = 38.98; *p* < 0.0001], *Per3* [F(1, 72) = 8.18; *p* < 0.01], and *Bmal1* [F(1, 68) = 10.81; *p* < 0.01] (Figure 5C,E,F). In the **SCN**, there was a main effect of ZT on *Per2* [F(1, 79) = 6.27; *p* < 0.05) and *Clock* [F(1, 79) = 4.08; *p* < 0.05], and a main effect of treatment on *Per2* expression [F(1, 79) = 6.27; *p* < 0.05]. Tukey’s post hoc analysis revealed that 40 mg/kg of apremilast increased SCN *Per2* expression. For the **VTA**, no significant effects were observed between treatment and ZT, nor were there any treatment x ZT interactions. However, there was a main effect of ZT on *Per2* [F(1, 49) = 13.36; *p* < 0.001] expression, whereby expression was found to be higher ZT15 (Figure 5K).

## 3. Discussion

By uncovering the complex bidirectional relationship between harmful alcohol use and the molecular clock, we hope to better inform optimal AUD treatments. Genetic variations in circadian genes have been associated with excessive alcohol use in humans [31]. Moreover, the harmful effects of alcohol misuse on circadian rhythmicity are thought to stem from alterations in the dopamine reward system [31,32,33]. Dopaminergic inputs from the VTA to the SCN are necessary for normal circadian entrainment [34]. ClockΔ19 (which have a dominant negative *Clock* mutation) have increased dopaminergic tone, altered dopamine receptor function, voluntarily consume more ethanol, and show higher ethanol preferences at relatively higher ethanol concentrations (18–21%) compared to their wild-type littermates (mixed BALBc/C57BL/6J) [12,35]. The consequences of chronic, harmful alcohol use on this system are magnified by the fact that circadian genes regulate the expression of thousands of downstream genes responsible for various functions, such as neurotransmission, immune signalling, and energy homeostasis–many of which are disrupted in AUD [36,37,38,39,40]. To this end, the present study sought to identify the effects of chronic binge-like ethanol on circadian gene expression across a network of circadian and dopaminergic-related brain regions-the NAc, VTA and SCN.

By characterizing the effects of chronic binge-like ethanol drinking on diurnal circadian gene expression patterns across these important brain regions, we hope to drive future consideration of chronotherapeutic drug therapy-timing drugs delivery with circadian rhythms-and the use of pharmacotherapies that target circadian gene expression (such as PF-670462 and apremilast) to treat AUDs (the basis of experiments 2 and 3).

Prior work addressing the role of central circadian genes in regulating alcohol intake have largely focused on ethanol intake and preference using two-bottle choice ethanol drinking. A key concern for this method is the inability to discern whether rodents consume pharmacologically intoxicating ethanol levels. Therefore, we elected to use the widely adopted drinking in the dark (DID) task, whereby limited access to ethanol (typically 10–20%) is offered 3 h after lights off (during their active cycle [Zeitgeber Time (ZT 15)]. This reliable measure of binge-like drinking results in intoxicating blood alcohol levels (BALs) across high drinking strains [29,30]. 

In experiment 1, chronic binge-like ethanol drinking reduced the expression of several NAc and SCN circadian genes in female HDID-1 mice. These results align with the work of Huang, Ho [41], which found a reduction in peripheral *Clock* gene expression in individuals with an AUD diagnosis, and Chen, Kuhn [42], which reported that 2 weeks of ethanol-feeding changed the phase of *Per2* and *Per3* expression in the SCN of male rats. In support, genetic mutations in any of the three *Per* genes has been associated with increased alcohol intake [7,43,44,45,46]. Although we saw no effects of chronic binge-like ethanol drinking on the phase of any of the circadian genes tested, we did see a reduction in the amplitude of SCN *Per2* and *Per3* expression in female HDID-1 mice, suggesting that chronic alcohol intake may alter circadian gene patterns in the central circadian pacemaker. In support, RNAi mediated knockdown of VTA *Clock* expression resulted in an increase ethanol intake [47,48]. This work highlights a likely importance of VTA circadian function in gating harmful alcohol use (in male animals); however, we saw no effect of binge-like ethanol drinking on the expression of any circadian genes in the VTA of female HDID-1 mice. Importantly, future work should also consider the role of post-translational modification and protein activity (i.e., protein phosphorylation) of circadian genes in response to harmful alcohol drinking.

In contrast to the present findings, a recent study found that 4 days of binge-like ethanol drinking increased *Clock* expression in the NAc shell sub-region and that antisense mediated *Clock* knockdown reduced ethanol intake in in male C57BL/6J mice, suggesting that a maladaptive increase in NAc circadian genes may contribute to harmful drinking [14]. Moreover, this study found no effect of NAc Clock knockdown on water or sucrose intake, suggesting that the observed reduction in intake was specific to alcohol and not a general decrease in liquid intake or an effect on reward processing/malaise [14]. Interestingly, these findings on acute binge-like drinking are in clear contrast to our own, wherein chronic binge-like drinking reduced circadian gene amplitude in the NAc and SCN (but not the VTA). This suggests that circadian gene responses to alcohol may change over time and raises the possibility that a reduction in circadian gene expression following chronic alcohol intake may be compensatory [12].

There are known sex differences in the effects of harmful alcohol intake on human sleep and other rhythmic processes [49,50]. However, the bulk of pre-clinical and clinical research on these topics and their underlying mechanisms have only been tested in male subjects. This is an especially important concern given the pronounced rise in human females diagnosed with an AUD and emerging evidence of sex differences in the physiological, neural and psychological consequences of harmful alcohol use [51]. In response, female HDID-1 mice were chosen to observe the effects of chronic binge-like ethanol drinking circadian gene expression across a network circadian and reward-related brain regions, many of which (such as the NAc, VTA and SCN) are considered sexually dimorphic [52,53,54]. Moreover, there are clear sex differences in binge-like ethanol drinking in response to chemogenetic activation and inhibition of the NAc [25,55,56,57]. A limitation to experiment 1 is the lack of a male group to characterize differences across sexes. Future work on these topics (and all biomedical research) should test both sexes simultaneously. Circulating ovarian steroids are known to change sleep and circadian rhythms in women [58]; therefore, it follows that sex hormones may also have unique and interesting interactions across circadian gene expression, harmful alcohol use and sexually dimorphic brain regions/responses to chronic alcohol. Although we saw no sex differences in neural circadian gene expression among iHDID-1 mice treated with 0 mg/kg (experiment 3), circulating hormones were not accounted for in the present study and direct effect of female and male sex hormones (both of which fluctuate diurnally) on brain-wide circadian gene expression cannot be inferred [59]. Taken together, the role of sex, sexually dimorphic brain regions, and circadian-related mechanisms remains an important future direction.

We observed a pronounced circadian peak (ZT 15) and trough (ZT 3) in the NAc expression of *Per2* (with similar patterns for *Per1* and *Per3*). *Per2* is considered a reliable proxy for the overall circadian gene turnover [60], which supports its candidacy as a primary reporter gene in bioluminescent, luciferase-based reporter mice [61,62]. Considering our hypothesis that the *CK1e*/*d* inhibitor, PF-670462, and the PDE4 inhibitor, apremilast, would increase circadian gene expression following chronic harmful drinking, we decided to test the effects of these drugs at ZT 15, wherein the most pronounced ethanol-induced blunting of *Per2* expression was seen. This time also coincides with the start of DID, which was characterized as being the optimal timeframe to capture binge-like intake [63]. It would be important to consider whether the entrainment of bottle presentation at ZT 15 (which occurred identically for ethanol and water control mice) had any impact on circadian gene expression (especially for *Per2*).

Pharmacologically targeting casein-kinase-1 lengthens the period of the molecular clock and has gained recent attention as a promising therapeutic direction for a range of psychiatric and addictive disorders [17,32,64,65].The *CK1e*/*d* inhibitor, PF-670462, was shown to dose dependently reduce and prevent the ‘Alcohol Deprivation Effect’ in male rats, a chronic model of relapse-like drinking [18]. Here, Perreau-Lenz, Vengeliene [18] determined that systemic *CK1e*/*d* inhibition blunted the high daytime ethanol intake period emblematic of alcohol re-exposure in this model and induced a persistent phase shift towards daytime for both locomotor activity and saccharin intake. This suggests that *CK1e*/*d* inhibition may cause a desynchronization of rhythmic physiological functions, which may in turn contribute to a reduction in relapse-like drinking. In contrast, we saw no comparable effect of *CK1e*/*d* inhibition on binge-like ethanol drinking in female HDID-1 mice (experiment 2). A stark difference in drinking paradigm, species, and the fact that opposite sexes were tested make direct comparisons between these two studies notably difficult. *CK1e*/*d* functions to destabilize *Per2*, which is known to play an important role in reducing harmful alcohol drinking [66]. Although there was no direct effect of PF-670462 on *Per2* expression in any of the brain regions tested, we did observe a trending increase in the NAc expression of *Npas2*–an established regulator of the *Per* genes. Because we saw no effect of chronic binge-like drinking on circadian gene phase or period, these factors were not directly considered for the present experiment or its analysis. The effect of PF-670462 on behavioral measures of circadian phase and relapse-like drinking is in direct contrast to our own findings, whereby CK1e/d inhibition had no effect on ethanol intake. Future work should address circadian gene expression in models of harmful drinking across multiple timepoints (similar to experiment 1) and in several relevant drinking paradigms, species, and strains.

In experiment 3, apremilast was found to decrease binge-like ethanol intake but had no significant effect on rescuing circadian rhythms from the harmful effects of alcohol. Inhibiting PDE4, via rolipram, has been shown to increase the amplitude of *Per2* expression in cell culture [22]. Although rolipram and apremilast are both FDA approved selective PDE4 inhibitors, apremilast is preferred to rolipram as it is shown to have less gastrointestinal side effects [67]. Inhibiting PDE4 is shown to reduce ethanol preference and intake [23,25,27]. Similarly, we found inhibiting PDE4 with apremilast resulted in a significant decrease in binge-like ethanol intake [25]. A limitation of this experiment is the exclusion of a water control group, which was based on previous evidence that apremilast had no effect on saccharin or water intake in male and female HDID-1 mice [25], and to limit the number of animals used. Comparably, preclinical models show apremilast reduces ethanol intake but not water intake, in C57BL/6J female and male mice given a two-bottle choice drinking model [68]. For both experiments 2 and 3, we did observe a handling effect in response to i.p. injections and this limitation should be considered in future behavioral pharmacology studies addressing chronic binge-like drinking. Taken together, the dissonance between the effects of PF-670462, which changed circadian expression, but not ethanol intake, in contrast to apremilast, which reduced drinking despite no effects on circadian genes, highlights an important limitation in our understanding between the role of circadian gene expression and binge-like ethanol drinking.

The effects of harmful alcohol use on the hypothalamic-pituitary-adrenal system is well known and the role of circulating glucocorticoids in AUD development continues to draw attention as a viable mechanism for treatment [69]. Greater circadian misalignment (a shorter interval between melatonin onset and the mid-sleep point) is associated with higher alcohol intake [70]. Moreover, human cortisol levels fluctuate in a diurnal pattern, with levels peaking just after waking and tapering throughout the day. Harmful alcohol use and stress are both known to blunt this response [71,72]. For instance, individuals with an AUD who were alcohol abstinent showed a reduced cortisol awakening response [71]. DBA/2J mice similarly show a decrease in corticosterone amplitude in response to 15-weeks of alcohol consumption [73]. The selective glucocorticoid receptor (GR) antagonist CORT113176 was shown to reduce binge-like ethanol drinking in female and male HDID-1 mice, but not their heterogeneous founders, the HS/Npt [74]. Recent work by Savarese, Grigsby [75] found no major differences in basal corticosterone levels between HDID-1 and HS/Npt mice, levels were only compared at one timepoint (during the light cycle, when corticosterone levels are relatively low). It would be interesting to see whether differences in daily glucocorticoid oscillations between HDID-1 and HS/Npt mice exist and whether chronic drinking differentially impacts these rhythms.

Male HDID-1 mice are known to have shorter free-running periods than and more variable/fragmented daily activity patterns than male HS/Npt mice, whereas mice selectively bred for high severity of ethanol induced-convulsions [Withdrawal Seizure-Prone (WSP-2); a model of heightened withdrawal severity] displayed longer free-running periods compared to Withdrawal Seizure Resistant-2 mice [76]. Moreover, this suggests that HDID-1 and iHDID-1 may have important baseline differences in brain-wide circadian gene expression patterns and may respond differently to harmful alcohol intake compared to other strains. Future work looking at the interface of stress and alcohol should also consider the importance of testing multiple timepoints (such as the eight timepoints tested in the present study) and the use of unique genetic models of ethanol-related behaviors with known differences in circadian phenotypes.

## 4. Methods and Materials

### 4.1. Animals

Experiments 1 and 2 used adult female HDID-1 mice (S29.G31 and S33.G35), which were selectively bred to reach high blood alcohol levels (BALs) in the limited access, Drinking in the Dark (DID) test [77,78]. These mice reliably reach BALs over 100 mg% [whereby >80 mg% (equivalent to 80 mg/dL)] is considered intoxicated and represent a unique genetic risk model for binge-like ethanol drinking [28]. For experiment 3, female and male inbred HDID-1 (iHDID-1; S26:G21-24) mice were used, which were derived from HDID-1 mice (S25) and similarly show stable high levels of binge-like ethanol drinking and reliably reach intoxicating BALs [79]. All mice were bred and maintained in the Veterans Affairs Portland Health Care System Veterinary Medical Unit, on a reverse 12-h/12-h light/dark schedule, with lights off at 7:30 am (PST). Experimental rooms were maintained at a temperature of 21 ± 1 °C. Purina 5LOD chow (PMI Nutrition International, Brentwood, MO, USA) was available ad libitum. Mice were housed in standard polycarbonate cages with stainless steel wire tops on Bed-o’cobs^®^ bedding (The Andersons, Inc., Maumee, OH, USA), and were habituated to single housing conditions for 5–7 days prior to experiments. All procedures were approved by the local Institutional Animal Care and Use Committee and were conducted in accordance with NIH Guidelines for the Care and Use of Laboratory Animals.

### 4.2. Drugs

Ethanol (200 proof, Decon Labs, King of Prussia, PA, USA) was dissolved in tap water to a 20% *v*/*v* ethanol solution for all experiments. The CK1 e/d inhibitor, PF-670462 (GlaxoSmithKline, Verona, Italy; Experiment 2), and the PDE4 inhibitor, apremilast (Toronto Research Chemicals, Ontario, Canada; Experiment 3) were administered intraperitoneally (i.p.) injection in volume of 10 mL/kg per mouse body weight. For experiment 2, mice were treated with 30 mg/kg of PF-670462 or 0 mg/kg (sterile saline). In experiment 3, mice were treated with apremilast (40 mg/kg) or 0 mg/kg [Tween-80 (1.75% *v*/*v* in sterile saline)] 30–60 min prior to DID.

### 4.3. Experiment 1: Determining Effects of Chronic Binge-like Drinking on Circadian Gene Expression in Brain Regions Important for AUDs

To determine potential circadian gene changes in response to chronic binge-like drinking, female HDID-1 mice (S29.G31) were given voluntary access to 20% ethanol (*v*/*v*, in tap water) or water in a 4-day DID for 8 weeks. There is a dearth of data for female subjects in neuroscience research, representing a major gap in the field. For this reason, we elected to fill this gap by using only females in this experiment (a limitation expanded upon in the discussion). Each week, a single bottle of 20% ethanol or water was provided 3-h into the dark cycle for 2-h (from ZT 15-17) for days 1–3 and 4-h (ZT 15-19) on day 4 (n = 192 mice; where n = 6–7 mice/fluid type/time point). No ethanol was offered on days 5–7 and mice had unlimited access to water (not measured). The drinking paradigm continued for 8 weeks to ensure chronic binge-like drinking and develop a stable drinking pattern. The sipper tubes were weighed before and after each DID session to calculate fluid consumption. Mice were weighed 3–5 h after DID on days 1 and 3, for all 8 weeks. The amount of ethanol intake (g/kg) or water intake (mL/kg) was calculated daily.

To measure rhythmic gene expression across a 24-h period, groups of mice were euthanized at 3-h intervals starting 21-h after the last DID access [n = 192 mice; where n = 6–7 mice per fluid type (water or ethanol) per time point (8 total; ZT 0, 3, 6, 9, 12, 15, 18, 21)]. Whole brains were dissected, flash frozen on dry ice, and sectioned (at 300 µm) on a cryostat. Brain sections were stored at −80 °C prior to processing. Frozen tissue was collected using a 0.5 mm (SCN and VTA) and 1 mm (NAc) tissue puncher and processed for qPCR (see below).

### 4.4. Experiment 2: Determine Whether Inhibition of CK1e/d (via Administration of PF-670462) Reduces Ethanol Intake and Rescues Ethanol-Induced Circadian Gene Disruptions

It was previously shown that the *CK1e*/*d* inhibitor, PF-670462, reduced relapse-like drinking in male Wistar rats in a dose dependent manner (0, 15, and 30 mg/kg). To build from this work, we elected to test whether PF-670462 (0 or 30 mg/kg) would reduce chronic (8 weeks) binge-like ethanol intake in female HDID-1 mice (S33.G35; as described above for experiment 1). In weeks 7 and 8, mice received an i.p. injection of saline or PF-670462 (30 mg/kg) 30–60 min prior to each DID session (n = 26 mice; where n = 5–7/sex/time point). In Experiment 1, the timepoints ZT3 and ZT15 were identified to be a trough and peak in *Per2* circadian gene expression, respectively. Therefore, tissue (NAc, VTA, and SCN) was collected one day after the last DID at these two time points.

### 4.5. Experiment 3: Determine Whether Inhibition of PDE4 (via Administration Apremilast) Reduces Ethanol Intake and Rescues Ethanol-Induced Circadian Gene Disruptions

Male and female HDID-1 mice were subjected to 8 weeks of an ethanol DID (as described above for experiment 1), wherein mice received an i.p. injection of 0 mg/kg or apremilast (40 mg/kg) 30–60 min prior to each DID session on weeks 7 and 8 (n = 76; where n = 9–12/sex/time point). As for Experiment 2, NAc, VTA and SCN tissue was collected at ZT 3 and 15 for qPCR analysis. No water group was included in this experiment for the following reasons: 1) alcohol effects on circadian gene expression have been observed in both sexes in several studies and have required hundreds of mice (time points necessitate large numbers of animals) and we aim to comply with the three R’s of animal research (*reduce*, refine, replace) by reducing the number of mice used in studies where possible, and 2) our previous work and others work found no effects of apremilast on water or saccharin intake in HDID-1, HDID-2, or C57BL/6J mice [25].

### 4.6. Quantitative Real-Time Polymerase Chain Reaction (qPCR)

RNA isolation and cDNA synthesis was performed according to a previously established method [75]. Following tissue collection of the NAc, SCN, and VTA, RNA was isolated in Purezol reagent and collected using the Bio-Rad Aurum Total RNA Fatty and Fibrous Tissue Kit. RNA was reverse transcribed into cDNA using the Bio-Rad iScript cDNA Synthesis Kit. RT-qPCR reactions were carried out in duplicate with negative controls (primer only controls, and no reverse transcriptase sample controls) using the Bio-Rad SYBER Green Supermix (Bio-Rad, Hercules, CA, USA), gene-specific primers, and the Bio-Rad CFX384 Real-Time System. Established primers targeting *Clock*, *Npas2*, *Bmal1*, *Per1*, *Per2*, *Per3*, and *18s* (primer sequences provided in Table 2) [9,12,80].

### 4.7. Statistical Analysis

The qPCR data were analyzed using the ΔΔCt method for determining relative gene expression [80]. Behavioral and molecular data were analyzed using a Student’s *t*-test and a two-way analysis of variance (ANOVA; treatment × ZT) in Experiments 1 and 2. In Experiment 3, a three-way ANOVA (sex × treatment × ZT) is used for each gene and across brain regions. When no significant sex × drug interaction was observed, data were collapsed across sex for analyses. Analyses were performed and graphs were prepared using GraphPad Prism ver.9 (GraphPad Software, San Diego, CA, USA). All values are presented as mean ± standard error (SE) and significance set at an alpha value of 0.05.

## Figures and Tables

**Figure 1 ijms-23-11084-f001:**
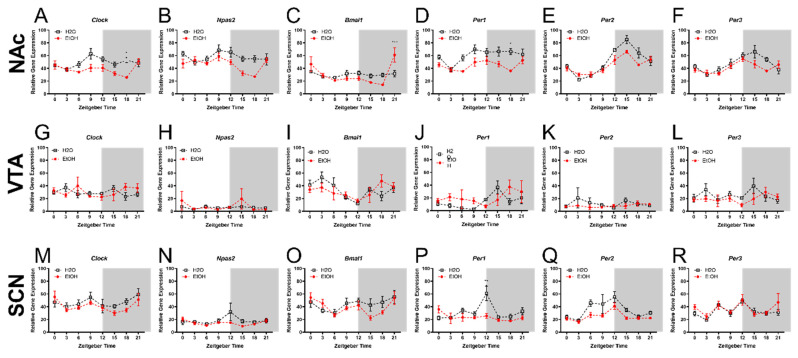
Chronic binge-like ethanol intake reduces NAc and SCN expression of circadian genes in female HDID-1 mice. Detailed descriptive statistics are provided in Table 1. The following core circadian genes—*Clock*, *Npas2*, *Bmal1*, *Per1*, *Per2*, *Per3*—were analyzed across the NAc (**A**–**F**); VTA (**G**–**L**); and SCN (**M**–**R**). (**A**) *Clock*—Main effect of fluid type [F(1, 80) = 21.61; *p* < 0.0001] and ZT [F(7, 80) = 2.72; *p* < 0.01]. (**B**) *Npas2*—Main effect of fluid type [F(1, 80) = 18.01; *p* < 0.0001] and ZT [F(7, 80) = 3.22; *p* < 0.001]. (**C**) *Bmal1*—Main effect of ZT [F(7, 80) = 7.33; *p* < 0.0001] and an interaction between fluid x ZT [F(7, 80) = 4.92; *p* < 0.001]. (**D**) *Per1*—Main effect on fluid type [F(1, 80) = 34.67; *p* < 0.0001] and ZT [F(7, 80) = 3.72; *p* < 0.001]. (**E**) *Per2*—Main effect on fluid type [F(1, 80) = 7.03; *p* < 0.01] and ZT [F(7, 80) = 27.64; *p* < 0.0001] and a fluid type x ZT interaction [F(7, 80) = 2.74; *p* < 0.01]. (**F**) *Per3*—Main effect on fluid type [F(1, 80) = 5.28; *p* < 0.05] and ZT [F(7, 80) = 6.68; *p* < 0.0001]. (**G**) Clock—No main effect or interaction. (**H**) *Npas2*—No main effect or interaction. (**I**). *Bmal1*—No main effect or interaction. (**J**) *Per1*—No main effect or interaction. (**K**) *Per2*—No main effect or interaction. (**L**) *Per3*—No main effect or interaction. (**M**) *Clock*—Main effect of ZT [F(7, 76) = 0.48; *p* < 0.01]. (**N**) *Npas2*—No main effect or interaction. (**O**) *Bmal1*—Main effect of ZT [F(7, 76) = 3.15; *p* < 0.01]. (**P**) *Per1*—Main effect of fluid type [F(1, 74) = 9.08; *p* < 0.01] and ZT F(7, 74) = 3.96; *p* < 0.001]. (**Q**) *Per2*—Main effect of fluid type [F(1, 76) = 12.22; *p* < 0.001] and ZT [F(7, 76) = 5.87; *p* < 0.0001]. (**R**) *Per3*—Main effect of ZT [F(7, 80) = 4.98; *p* < 0.001].

**Figure 2 ijms-23-11084-f002:**
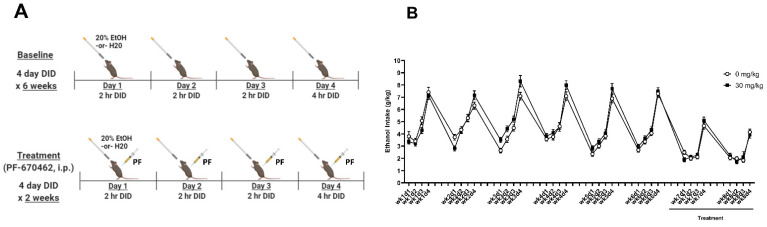
The CK1E/D inhibitor, PF-670462, did not reduce chronic binge-like ethanol intake in female HDID-1 mice. (**A**) Experimental timeline for baseline binge-like drinking (weeks 1–6) and PF-670462 treatment relative to DID (weeks 7 & 8). (**B**) Ethanol intake (g/kg) for Days 1–4 across 6-weeks with PF-670462 treatment administered in weeks 7 and 8. Average 2-h ethanol intake during Days 1–3 for weeks 7 and 8 revealed no main effects of treatment [F(1, 24) = 0.001], day [F(1, 24) = 0.70], or treatment x day interactions [F(1, 24) = 3.11]. Average 4-h ethanol intake on Day 4 during weeks 7 and 8 revealed no main effect of treatment.

**Figure 3 ijms-23-11084-f003:**
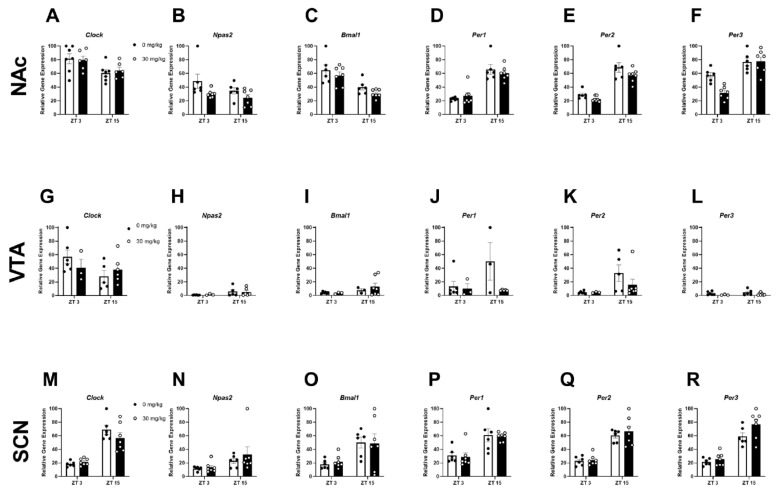
Inhibiting CK1E/D with PF-670462 treatment reduces NAc circadian gene expression in female HDID-1 mice. The following core circadian genes—*Clock*, *Npas2*, *Bmal1*, *Per1*, *Per2*, *Per3*—were analyzed across the NAc (**A**–**F**); VTA (**G**–**L**); and SCN (**M**–**R**). (**A**) *Clock*—Main effect of ZT [F(1, 22) = 9.39; *p* < 0.01]. (**B**) *Npas2*—Main effect of treatment [F(1, 22) = 5.81; *p* < 0.05]. (**C**) *Bmal1*—A main effect of ZT [F(1, 22) = 23.83; *p* < 0.0001]. (**D**) *Per1*—Main effect of ZT [F(1, 22) = 64.84; *p* < 0.0001. (**E**) *Per2*—Main effect of treatment [F(1, 22) = 4.94; *p* < 0.05] and ZT [F(1, 22) = 83.76; *p* < 0.0001]. (**F**) *Per3*—Main effect of treatment [F(1, 22) = 6.17; *p* < 0.05], ZT [F(1, 22) = 42.02; *p* < 0.0001], and a treatment x ZT interaction [F(1, 22) = 6.92; *p* < 0.05]. (**G**) *Clock*—No main effect or interaction. (**H**) *Npas2*—No main effect or interaction. (**I**) *Bmal1*—No main effect or interaction. (**J**) *Per1*—No main effect or interaction. (**K**) *Per2*—No main effect or interaction. (**L**) *Per3*—No main effect or interaction. (**M**) *Clock*—Main effect of ZT [F(1, 22) = 63.45; *p* < 0.0001]. (**N**) *Npas2*—Main effect of ZT [F(1, 22) = 5.17; *p* < 0.05]. (**O**) *Bmal1*—Main effect of ZT [F(1, 22) = 11.25; *p* < 0.01]. (**P**) *Per1*—Main effect of ZT [F(1, 22) = 24.80; *p* < 0.0001]. (**Q**) *Per2*—Main effect of ZT [F(1, 22) = 62.67; *p* < 0.0001]. (**R**) *Per3*—Main effect of ZT [F(1, 22) = 70.08; *p* < 0.0001].

**Figure 4 ijms-23-11084-f004:**
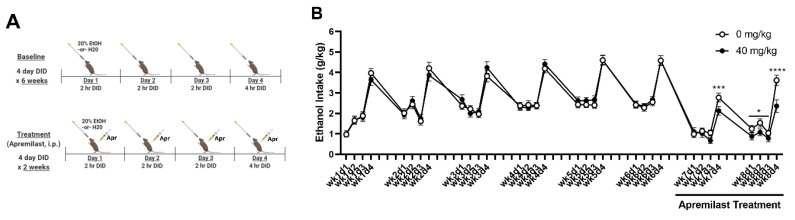
Treatment with the PDE4 inhibitor, apremilast, did reduce chronic binge-like ethanol intake in female and male iHDID-1 mice. (**A**) Experimental timeline for baseline binge-like ethanol drinking (weeks 1–6) and apremilast treatment relative to DID (weeks 7 & 8). (**B**) Ethanol intake (g/kg) for Days 1–4 across 6 weeks with apremilast treatment administered in weeks 7 and 8. Main effect of treatment [F(3, 88) = 27.20; *p* < 0.0001] and ZT [F(1, 82) = 4.33; *p* < 0.05] during Days 1–3 during weeks 7 and 8. Average 4 h intake on Day 4 revealed a main effect of treatment [F(1, 82) = 53.61; *p* < 0.0001] and a treatment x time interaction [F(1, 82) = 7.08; *p* < 0.0001]. No main effects of sex; thus, data are collapsed across sex. * = *p* < 0.05; *** = *p* < 0.001; **** = *p* < 0.0001.

**Figure 5 ijms-23-11084-f005:**
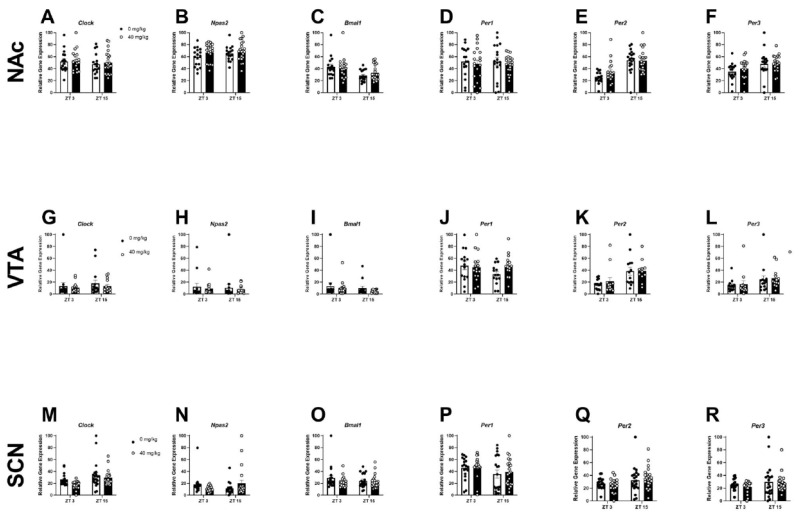
Apremilast treatment did not rescue the effects of chronic binge-like ethanol intake on circadian gene expression in male and female HDID-1 mice. The following core circadian genes—*Clock*, *Npas2*, *Bmal1*, *Per1*, *Per2*, *Per3*—were analyzed across the NAc (**A**–**F**); VTA (**G**–**L**); and SCN (**M**–**R**). (**A**) *Clock*—No main effect or interaction. (**B**) *Npas2*—No main effect or interaction. (**C**) *Bmal1*—Main effect of ZT [F(1, 68) = 10.81; *p* < 0.01]. (**D**) *Per1*—No main effect or interaction. (**E**) *Per2*—Main effect of ZT [F(1, 73) = 38.98; *p* < 0.0001]. (**F**) *Per3*—Main effect of ZT [F(1, 72) = 8.18; *p* < 0.01]. (**G**) *Clock*—No main effect or interaction. (**H**) *Npas2*—No main effect or interaction. (**I**) *Bmal1*—No main effect or interaction. (**J**) *Per1*—No main effect or interaction. (**K**) *Per2*—Main effect of ZT [F(1, 49).= 13.36; *p* < 0.001]. (**L**) *Per3*—No main effect or interaction. (**M**) *Clock*—Main effect of treatment [F(1, 79) = 9.81; *p* < 0.001] and ZT [F(1, 79) = 4.08; *p* < 0.05]. (**N**) *Npas2*—No main effect or interaction. (**O**) *Bmal1*—No main effect or interaction. (**P**) *Per1*—No main effect or interaction. (**Q**) *Per2*—Main effect of ZT [F(1, 79) = 6.27; *p* < 0.05]. (**R**) *Per3*—No main effect or interaction. No main effects of sex; thus, data are collapsed across sex.

**Table 1 ijms-23-11084-t001:** Chronic binge-like ethanol drinking reduces expression of circadian genes in the NAc and SCN of female HDID-1 mice in Experiment 1. Descriptive statistics for the NAc, VTA, and SCN.

Descriptive Statistics for Experiment 1							
**NAc**	**Main effects**	**Clock**	**Npas2**	**Bmal1**	**Per1**	**Per2**	**Per3**
	**Fluid Type**	F(1, 80) = 21.61	F(1, 80) = 18.01	F(1, 80) = 0.03	F(1, 80) = 34.67	F(1, 80) = 7.03	F(1, 80) = 5.28
	*p*-value	****	****	**ns**	****	**	*
	**ZT**	F(7, 80) = 2.72	F(7, 80) = 3.22	F(7, 80) = 7.33	F(7, 80) = 3.72	F(7, 80) = 27.64	F(7, 80) = 6.68
	*p*-value	*	**	****	**	****	****
	**Interaction**	F(7, 80) = 2.83	F(7, 80) = 1.73	F(7, 80) = 4.92	F(7, 80) = 1.37	F(7, 80) = 2.74	F(7, 80) = 1.73
	*p*-value	*	**ns**	***	**ns**	*	**ns**
**VTA**	**Main effects**	**Clock**	**Npas2**	**Bmal1**	**Per1**	**Per2**	**Per3**
	**Fluid Type**	F(1, 74) = 0.10	F(1, 74) = 0.25	F(1, 74) = 0.08	F(1,70) = 1.68	F(1,73) = 1.11	F(1, 74) = 1.85
	*p*-value	**ns**	**ns**	**ns**	**ns**	**ns**	**ns**
	**ZT**	F(7, 74) = 0.35	F(7, 74) = 0.86	F(7, 74) = 1.67	F(7,70) = 1.62	F(7,73) = 0.44	F(7, 74) = 0.85
	*p*-value	**ns**	**ns**	**ns**	**ns**	**ns**	**ns**
	**Interaction**	F(7, 74) = 1.47	F(7, 74) = 0.50	F(7, 74) = 1.22	F(7,70) = 1.62	F(7,73) = 0.44	F(7, 74) = 0.89
	*p*-value	**ns**	**ns**	**ns**	**ns**	**ns**	**ns**
**SCN**	**Main effects**	**Clock**	**Npas2**	**Bmal1**	**Per1**	**Per2**	**Per3**
	**Fluid Type**	F(1, 76) = 2.78	F(1, 74) = 2,74	F(1, 76) = 1.71	F(1, 74) = 9.08	F(1, 76) = 12.22	F(1, 75) = 2.08
	*p*-value	**ns**	**ns**	**ns**	**	***	**ns**
	**ZT**	F(7, 76) = 2.27	F(7, 74) = 1.32	F(7, 76) = 3.15	F(7, 74) = 3.96	F(7, 76) = 5.87	F(7, 80) = 4.98
	*p*-value	*	**ns**	**	***	****	***
	**Interaction**	F(7, 76) = 0.48	F(7, 76) = 0.93	F(7, 80) = 1.01	F(7, 74) = 3.66	F(7, 76) = 0.84	F(7, 80) = 0.78
	*p*-value	**ns**	**ns**	**ns**	**	**ns**	**ns**

ns = nonsignificant; * = *p* < 0.05; ** = *p* < 0.01; *** = *p* < 0.001; **** < 0.0001.

**Table 2 ijms-23-11084-t002:** Primer sequence for circadian gene expression analyzed by qPCR for the NAc, SCN, and VTA.

Gene	Forward (5′-3′)	Reverse (5′-3′)
*18s*	ACCGCAGCTAGGAATAATGGA	GCCTCAGTTCCGAAAACCA
*Clock*	CAGAACAGTACCCAGAGTGCT	CACCACCTGACCCATAAGCAT
*Npas2*	GACACTGGAGTCCAGACGCAA	AATGTATACAGGGTGCGCCAAA
*Bmal1*	CCAAGAAAGTATGGACACAGACAAA	GCATTCTTGATCCTTCCTTGGT
*Per1*	CTCTGTGCTGAAGCAAGACCG	TCATCAGAGTGGCCAGGATCTT
*Per2*	GAGTGTGTGCAGCGGCTTAG	GTAGGGTGTCATGCGGAAGG
*Per3*	GTCCATCTGGAGAATGATAGAGCG	GCTTCAGCACCTCCTCTCGAC

## Data Availability

The gene expression and ethanol intake data supporting the present findings have been deposited and are available in the Figshare digital repository (10.6084/m9.figshare.21161371).

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
