# Peer review of "Targeting the Maladaptive Effects of Binge Drinking on Circadian Gene Expression"

_ijms, 2022, doi:10.3390/ijms231911084_

Round 1

Reviewer 1 Report

This is a well done work that investigate the relationship between circadian rhythms and alcohol abuse. The protocol and the results are interesting and the discussion is very well based. Although the study demonstrates many limitations their are also well discussed.

Overall a very good and interesting work and a valuable contribution to the scientific literature; although the results may be interesting, the are some concerns that should be addressed:

GENERAL COMMENTS:

Why 0mg/kg is sometimes used and sometimes vehicle? I suggest to use just a single pattern to maintain the text clearer;

  All figures should be revised in terms of layout and also reconsidered if some could not be replaced by tables; 

SPECIFIC: 

LINES 30-33: The cited articles (3, 4 and 5) are not related to binge use, but to the effects of alcohol in according to time of the day. More appropriate references should be used or rephrase the sentence.

LINES 219-222: The graphics in figure 1 are showed in a low resolution and is not possible to take a good visualization, in addition, it is also possible to note that some figures have a symbol that is not described in the legend

LINE 257: I suggest insert PF-670462 (or PF) on top of the second syringe. Also considerer to Figure 4A.

LINE 278: The follow sentence is correct? This is not related to PF-670462 effect?  "Post-hoc analysis revealed that Per3 expression was significantly reduced in ethanol-treated mice compared to the water-treated control group at ZT 3 (Fig. 3E)."

LINE 287: The graphics in figure 3 are above their titles (A, B, C...).

LINE 312: (Fig. 2B) is correct?

LINES 313-314: Does this difference also occur in week 7? Because it's not apparent in  the figure

LINE 320: There is no difference in some weeks before the treatment? In particular, week 5 and 6 apparently have this difference in almost all the points

LINE 343: The graphics in figure 5 are over their titles (A, B, C...). In addition, the graphics are in a low resolution and could be a larger size and with less space between them

LINES 335-343: This description is confusing because the order of the citation is altered (and not always used) and sometimes it is described the brain structure and sometimes it's not.

LINES 368-372: This is correct? "all of which" are disrupted? The same to sex dimorphism (LINE 423)

LINES 372: The reference from Takahashi it's not related to AUD. The citation should be repositioned in the phrase

LINE 384: cite references to "widely adopted"

LINE 434: "iHDID-1" is correct?

Author Response

IJMS Reviewer comments

Reviewer 1:

GENERAL COMMENTS:

  1. Why 0mg/kg is sometimes used and sometimes vehicle? I suggest to use just a single pattern to maintain the text clearer.

-We agree that a consistency in how this is reported will increase the readability of this work. In response, we have decided to only use “0 mg/kg” throughout the entire manuscript.  

  1. All figures should be revised in terms of layout and also reconsidered if some could not be replaced by tables.

-We feel that in addressing the suggested edits to our figures (namely items 4, 5, and 7) we have greatly improved their aesthetic value and would like to thank the reviewer for their clear guidance and for helping improve this manuscript. While we agree with the reviewer that replacing some of the more unruly figure panels with tables may ease their dissemination, we argue that visually presenting individual data points is more scientifically meaningful (i.e., allows for a visual understanding of variability) than only providing the mean and SEM (or standard deviation). To clarify main effects, a table is provided for figure 1 that highlights the F- and p-values per 2-way comparison. Although a descriptive statistics table for the other gene expression figure panels (Figures 3 and 5) could be provided, we feel this is unwarranted given so few significant effects were seen in these experiments. Therefore, we elect to maintain the current number of tables.       

SPECIFIC: 

  1. LINES 30-33: The cited articles (3, 4 and 5) are not related to binge use, but to the effects of alcohol in according to time of the day. More appropriate references should be used or rephrase the sentence.

-Correct, these references are not directly related to binge drinking and we appreciate that the clarity of this sentence was brought to our attention. In response, we have more accurately reflected that “Among the health deficits associated with harmful alcohol use are a persistent disruption of the sleep/wake cycle [3] and disruptions in rhythmic physiological processes, such as body temperature and melatonin release [4, 5].”  

  1. LINES 219-222: The graphics in figure 1 are showed in a low resolution and is not possible to take a good visualization, in addition, it is also possible to note that some figures have a symbol that is not described in the legend.

-A few of the images were much lower quality than we had realized. Substantial effort was provided to ensure that 600 DPI was used throughout our figures.  

  1. LINE 257: I suggest insert PF-670462 (or PF) on top of the second syringe. Also considerer to Figure 4A.

- This was a very helpful suggestion, and we agree that these diagrams are much more understandable with these additions. We would again like to thank the reviewer for helping strengthen our work.

  1. LINE 278: The follow sentence is correct? This is not related to PF-670462 effect?  "Post-hoc analysis revealed that Per3 expression was significantly reduced in ethanol-treated mice compared to the water-treated control group at ZT 3 (Fig. 3E)."

-We apologize, this was supposed to be “Per2”, not “Per3”. We thank the reviewer for their careful reading. In response to this comment and that addressed in item 1, the following correction has been made to this sentence: “Post-hoc analysis revealed that Per2 expression was significantly reduced in ethanol-treated mice compared to the 0 mg/kg treated mice at ZT 3.”   

  1. LINE 287: The graphics in figure 3 are above their titles (A, B, C...).

-We are uncertain as to how the titles fell in the background for figure 3. This error has been corrected.  

  1. LINE 312: (Fig. 2B) is correct?

-Another helpful catch, this has been corrected to “4B”.

  1. LINES 313-314: Does this difference also occur in week 7? Because it's not apparent in the figure

-This difference does occur during the 4 hour drinking test on Day 4 of week 7, but not when comparing the 2-hour intake average across Days 1-3 of week 7 (KG). This is an important statistical distinction, and we highlighted this point in the results and discussion sections.

  1. LINE 320: There is no difference in some weeks before the treatment? In particular, week 5 and 6 apparently have this difference in almost all the points

-We greatly appreciate that this error was brought to our attention, as the wrong graph was provided here. There was in fact no group differences in ethanol drinking for any of the weeks prior to treatment, which was correctly reported. We have updated Figure 4B to reflect this. Again, we greatly appreciate the careful review provided here.    

  1. LINE 343: The graphics in figure 5 are over their titles (A, B, C...). In addition, the graphics are in a low resolution and could be a larger size and with less space between them.

-As mentioned in the response to item 2, care and attention was provided in following this comment and other image-related edit suggestions. In particular, the titles in figures 3 and 5 were brought to the foreground, given appropriate space, and exported at a higher resolution (6000 DPI).

  1. LINES 335-343: This description is confusing because the order of the citation is alter ed (and not always used) and sometimes it is described the brain structure and sometimes it's not.

-The authors appreciate the guidance for from this reviewer regarding the best way to report our results. In response, we have bolded each brain region per comparison as a means of helping the overall readability of this section.  

  1. LINES 368-372: This is correct? "all of which" are disrupted? The same to sex dimorphism (LINE 423)

-In agreement with the reviewer, the use of “all” in both of these sentences is perhaps an overstatement. We have qualified both of these sentences by suggesting that “many” of the downstream effects of harmful alcohol use (namely its effects on neurotransmission, immune signalling, and energy homeostasis – for which further references have been provided) and that “many” of the brain regions important for circadian and ethanol drinking behaviors (such as the NAc, VTA, and SCN) are considered sexually dimorphic.   

  1. LINES 372: The reference from Takahashi it's not related to AUD. The citation should be repositioned in the phrase.

-The provided reference was not related to AUD or binge-like drinking. We have now included references that support this claim.

  1. LINE 384: cite references to "widely adopted"

-Additional references have been provided.

  1. LINE 434: "iHDID-1" is correct? –

-The use of iHDID-1 here is correct. The inbred HDID-1 (iHDID-1) mouse line was used for Experiment 3, whereas non-inbred HDID-1 mice were used for experiments 1 and 2. We agree that this distinction could be better highlighted in our manuscript to mitigate any potential confusion.

Reviewer 2 Report

Overall Summary

These experiments investigate the relationship between alcohol consumption and genes critical to the regulation of circadian rhythm. The initial experiment considered the effect of alcohol consumption using the DID, single bottle exposure protocol, on gene expression evaluated via rt-PCR. Numerous effects on critical circadian rhythm genes were observed in the NAC and SCN at multiple time points with most effects occurring at 21hrs post final drinking session (ZT 21) and additional effects at ZT3 (27hrs post final drinking session). Two additional experiments were conducted utilizing pharmaceutical pretreatment of PF-670462 (CK1 e/d inhibitor) or apremilast (PDE4 inhibitor) to determine the mechanistic role of circadian rhythm targets in the regulation of binge-like alcohol drinking. These experiments showed little to no effect on the regulation binge-like alcohol intake, and had minimal effects on the genes regulating circadian rhythm in the NAC, VTA, and SNC. While minimal effects were observed, this research contributes to the overall literature regarding the interaction of alcohol drinking and circadian disruption. There is strong justification for continued pursuit of this topic and the null effects are equally informative.  

Page: 1

Line 11    Grammatical error

Line 13    Grammatical error

Line 16    This sentence doesn't make sense - you are testing genes or circadian rhythms? Or maybe you mean genes regulating circadian rhythms? Circadian rhythm genes? Very unclear. Are you suggesting there is an underlying difference in genes that poses an increased risk to binge alcohol or that alcohol binging

changes the expression of these circadian rhythm genes?

Line 18    You do not need parentheses

Line 24    The abstract needs major revisions to clearly communicate the goals and findings of the research. The abstract also needs to clearly acknowledge the pre-clinical model used - that information should not be buried in the methods.

Line 43 PERIOD needs to be all caps

Page: 2

Line 52    "In contrast" would be more appropriate given the disparity of the reports

Line 66    These should just be sentences, not numbered points

Line 74    This should be the lead into your final sentence of the introduction

Line 76    How is cAMP signaling part of the circadian molecular pathway?

Line 94    This paragraph does not provide a clear and direct review of the gene targets discussed. There needs to be a statement that sets up the purpose of the paragraph as simply stating that cAMP signaling is important is AUD is vague. It seems your goal is to communicate there are potential pharmaceutical targets of circadian rhythm genes that may also regulate alcohol drinking. However, this is not ever actually stated.

Line 97    I would not agree that a genetically modified strain of mice that binge drink ethanol at higher rates than C57 mice are a particularly translational model. They do serve as an important tool for AUD research, but not because they are especially generalizable to wild type mice or necessarily humans who binge drink.

Page: 3

Line 107  mg/dl

Line 128  Please move administration times to the specific experiment procedures

Line 130 Please list what the Tween-80 is dissolved in

Line 131  Were mice habituated to handling for injections for all experiments? Results indicate a handling effect.

Line 134  This is awkward here. I would prefer the methods simply list the use of female subjects and the limitations be addressed in the discussion only.

Page: 4

Line 171 By this logic, if alcohol drinking so reliably changes circadian rhythm genes, you do not need to complete experiment 1, or use water controls for experiment 2. Why is experiment 3 the exception where you have chosen to not use an experimental design with water controls?

Line 182 Grammatical error, check your verb tense

Page: 5

Line 198 An acknowledgment of the amount of fluid (water or ethanol) consumed is necessary, even if it is just in text (no graph).

Line 198 Please separate brain region analyses for each experiment or provide a distinct heading within the paragraph (bolded, underlined, etc) so the switch is quickly and easily noticed

Line 198 All interactions need to also list the type of post hoc test used (this is included in some places but not others)

Line 201 No need to repeat methods, only state methods. This is applicable to ALL results sections.

Line 203 You need to state the statistical test (factorial ANOVA) and in this case the factors and levels. This is necessary for ALL statistics.

Line 208 As these are all separate ANOVA's, it is not appropriate to combine the F values like this. Instead, indicate significant interaction and refer to table (rather than list all F and p values).

Line 210 No effect listed for Bmal1, but it is labeled on the graph as sig increase

Line 219 You should consider expanding your table to list post hoc comparisons

Line 219 Graphs are nearly impossible to read bc they are too small. It seems like not all post hoc comparisons are labeled - however, it might just be too blurry.

Line 219 Consider reorganizing the graph to be hours post drinking with the ZTs as my understanding is your brain samples are only taken starting at ZT 15 (according to Line 146)

Page: 6

Line 223 Figure captions should not repeat reporting of the statistics, ie the F and p values. Simply describe the graph and state the outcome. At the end define the * meaning(s) for all graphs; this should be consistent throughout, ie p<.05 is always *, p<.01 is always **, etc. This lengthy of a figure caption is difficult to read. Also, please bold the graph letter, ie (A)

Line 243 Please list ALL exact, p values. You can keep * and ns, but after the listed p-value, ie p=.21, ns or p=.01, **

This is important so your statistics can be easily validated by the reader

Page: 7

Line 254 List type of analysis including factors and levels

Line 257 You should include data for water intake as well as it could be affected by PF-670462

Line 267 Again, do not repeat methods or design; just state results and include statement of analysis type with factors and levels

Page: 8

Line 286 Statistical sig not indicated for interaction in graph

Page: 9

Line 302 Again, do not repeat methods or design; just state results and include statement of analysis type with factors and levels

Line 315 If you included the previous day 4 times, you would not see a drug effect on weeks 7 & 8 as those mice also show decreased drinking compared to controls despite them being treated the same up to week 7. By not stating the factors and levels it is unclear that you are not comparing all time points which is not appropriate.

Line 317 p value typo - should be p< .01 (after rounding)

Line 329 Again, do not repeat methods or design; just state results and include statement of analysis type with factors and levels

Line 339 It is not appropriate to include a post hoc test when there is not an interaction

Page: 11

Line 376 Consider you are measuring gene expression and not protein changes after treatment and this timeline may be different. You could also consider activity based changes in proteins (ie phosphorylation of your target sites)

Line 386 DID also has a two-bottle method and you are using a rodent line that has increased intake. Your phrasing should indicate a desire to maximize consumption rather than suggesting the other models do not reach physiological levels of intake (which they often do)

Line 391 Is this a dependence model? Note the contrast to binge

Line 398 This transition is clunky. Consider starting with the current findings instead of previously published work.

Line 403 Again, when discussing specific findings, please start with the findings from the current studies and provide supporting or contrasting citations as a follow up. It is fine to establish a topic with widely accepted statement supported by literature (like in the next paragraph) but it's awkward to start a discussion paragraph relaying specific findings from a different paper not from the current lab.

Author Response

IJMS Reviewer comments

Reviewer 2:

Overall Summary

These experiments investigate the relationship between alcohol consumption and genes critical to the regulation of circadian rhythm. The initial experiment considered the effect of alcohol consumption using the DID, single bottle exposure protocol, on gene expression evaluated via rt-PCR. Numerous effects on critical circadian rhythm genes were observed in the NAC and SCN at multiple time points with most effects occurring at 21hrs post final drinking session (ZT 21) and additional effects at ZT3 (27hrs post final drinking session). Two additional experiments were conducted utilizing pharmaceutical pretreatment of PF-670462 (CK1 e/d inhibitor) or apremilast (PDE4 inhibitor) to determine the mechanistic role of circadian rhythm targets in the regulation of binge-like alcohol drinking. These experiments showed little to no effect on the regulation binge-like alcohol intake, and had minimal effects on the genes regulating circadian rhythm in the NAC, VTA, and SNC. While minimal effects were observed, this research contributes to the overall literature regarding the interaction of alcohol drinking and circadian disruption. There is strong justification for continued pursuit of this topic and the null effects are equally informative.  

  1. Page 1 - Line 11    Grammatical error

-This error has been corrected.

  1. Line 13    Grammatical error

- This error has been corrected

  1. Line 16    This sentence doesn't make sense - you are testing genes or circadian rhythms? Or maybe you mean genes regulating circadian rhythms? Circadian rhythm genes? Very unclear. Are you suggesting there is an underlying difference in genes that poses an increased risk to binge alcohol or that alcohol binging changes the expression of these circadian rhythm genes?

-We agree with the reviewer that this sentence is perhaps unclear. In response, we have made the following change:

“Therefore, the present study sough to test the effects of chronic binge-like drinking on diurnal circadian gene expression patterns in the master circadian pacemaker (SCN), the ventral tegmental area (VTA), and the nucleus accumbens (NAc) in High Drinking in the Dark-1 (HDID-1) mice, a unique genetic risk model for drinking to intoxication.”

  1. Line 18    You do not need parentheses.

-This correction has been made.  

  1. Line 24    The abstract needs major revisions to clearly communicate the goals and findings of the research. The abstract also needs to clearly acknowledge the pre-clinical model used - that information should not be buried in the methods.

-In response to other items we feel we have adequately revised the abstract and we thank the reviewer for helping strengthen our manuscript.

  1. Line 43 PERIOD needs to be all caps

--This correction has been made. 

  1. Page: 2 - Line 52    "In contrast" would be more appropriate given the disparity of the reports.

--This correction has been made. 

  1. Line 66    These should just be sentences, not numbered points.

-These numbers have been removed.

  1. Line 74    This should be the lead into your final sentence of the introduction.

-We agree that this is better served as the lead-in for the final paragraph and has been moved accordingly. We thank the reviewer for helping strengthen our introduction.

  1. Line 76    How is cAMP signaling part of the circadian molecular pathway?

  1. Line 94    This paragraph does not provide a clear and direct review of the gene targets discussed. There needs to be a statement that sets up the purpose of the paragraph as simply stating that cAMP signaling is important is AUD is vague. It seems your goal is to communicate there are potential pharmaceutical targets of circadian rhythm genes that may also regulate alcohol drinking. However, this is not ever actually stated.

-Items 10 and 11 are being addressed together. We agree with the reviewer that the provided evidence for the importance of cAMP as a circadian-related pharmacotherapy to treat AUD was lacking. To address this concern, the following sentences were added and addition literature support was provided.

“Ethanol-induced neuroadaptations in circadian and other related molecular pathways are thought to be critical in AUD progression. Therefore, pharmacologically targeting circadian-related signaling to reduce harmful alcohol use may be a viable treatment option for AUDs…There is an established role of cAMP activity in gating circadian gene amplitude and rhythmicity in the suprachiasmatic nucleus (SCN), the master circadian regulator [20]. Moreover, cAMP-dependent signaling is considered a core component of the mammalian circadian pacemaker [22]. PDE4 is a known a regulator of cAMP and pharmacological inhibition of PDE4 inhibitor (rolipram) has been shown to increase rhythmic expression of Per2 in cell culture [21]”  

  1. Line 97    I would not agree that a genetically modified strain of mice that binge drink ethanol at higher rates than C57 mice are a particularly translational model. They do serve as an important tool for AUD research, but not because they are especially generalizable to wild type mice or necessarily humans who binge drink.

-In response to item 8 from reviewer 3, we feel we have adequately supported (with appropriate references) our case that the HDID-1 model has high “construct (maintains descriptive power for human AUD, i.e., accurately captures drinking to pharmacological intoxicating), face (resembles human binge drinking), and predictive (used to accurately predict human binge drinking outcomes) validity”

  1. Page: 3 - Line 107  mg/dl

-Although these values are valid and numerically equivalent, we agree with the reviewer that “mg/dl” is a more common way of presenting blood alcohol levels. This detail has been provided.

  1. Line 128  Please move administration times to the specific experiment procedures.

-These have been removed.

  1. Line 130 Please list what the Tween-80 is dissolved in.

-The detail that the Tween-80 is dissolved in sterile saline has been provided.

  1. Line 131  Were mice habituated to handling for injections for all experiments? Results indicate a handling effect.

-Mice were not habituated to injections prior to testing and upon additional post hoc analysis it was shown that the treatment weeks (7 and 8) for both experiments 2 and 3 were lower across treatment groups compared to pretreatment drinking. These details are now provided in the results section and this limitation is highlighted in the discussion section. We appreciate that our data was so well considered and we appreciate the thoughtfulness of this reviewer and for their help in improving our work. 

  1. Line 134  This is awkward here. I would prefer the methods simply list the use of female subjects and the limitations be addressed in the discussion only.

-We do not find this sentence any less awkward than the complete lack of justification for choosing only male subjects across most of the past century of biomedical research. On these grounds, we elect to keep this current sentence in the methods section.

  1. Page: 4 - Line 171 By this logic, if alcohol drinking so reliably changes circadian rhythm genes, you do not need to complete experiment 1, or use water controls for experiment 2. Why is experiment 3 the exception where you have chosen to not use an experimental design with water controls?

-The point of this sentence was to justify why no water controls were used in experiment 3. Drinking does change circadian rhythm genes, but there remains an incomplete understanding of the effects of chronic binge-like ethanol drinking on circadian genes. Moreover, the role of harmful drinking on circadian gene rhythmicity requires testing across ethanol drinking behaviors (i.e., dependence modes versus preference drinking), genetic backgrounds, species, and in both sexes.

  1. Line 182 Grammatical error, check your verb tense.

-This correction was made.

  1. Page: 5 - Line 198 An acknowledgment of the amount of fluid (water or ethanol) consumed is necessary, even if it is just in text (no graph).

-We agree that this should be included. The following sentence is now included in the results section: “The average 4-hr ethanol intake for weeks 1-8 was 4.43 ± 0.21 g/kg and the average 4-hr water intake over these weeks was  60.22 ± 2.02 mL/kg (data not shown).”

  1. Line 198 Please separate brain region analyses for each experiment or provide a distinct heading within the paragraph (bolded, underlined, etc) so the switch is quickly and easily noticed.

-To help the readability of our results section, the brain regions have been bolded for each comparison.

  1. Line 198 All interactions need to also list the type of post hoc test used (this is included in some places but not others)

-Effort and care was given in ensuring that the appropriate post-hoc testing is provided for each comparison in the results section.

  1. Line 201 No need to repeat methods, only state methods. This is applicable to ALL results sections.

-We feel reiterating the purpose of each experiment is valuable to the reader and we wish to keep these sentences in the results section.

  1. Line 203 You need to state the statistical test (factorial ANOVA) and in this case the factors and levels. This is necessary for ALL statistics.

  • Although the specifics of which statistical testing was used under what conditions/comparisons are provided in the “statistical analysis” section of the methods, we have added these details to the results section in response to this comment.

  1. Line 208 As these are all separate ANOVA's, it is not appropriate to combine the F values like this. Instead, indicate significant interaction and refer to table (rather than list all F and p values).

-This correction has been made

  1. Line 210 No effect listed for Bmal1, but it is labeled on the graph as sig increase.

-There was a treatment x time interaction, whereby ethanol increased Bmal1 expression only during ZT 21. This result is now provided.

  1. Line 219 You should consider expanding your table to list post hoc comparisons.

-Adding these details to table 1 greatly reduce its readability and we argue that providing main effects is sufficient here. 

  1. Line 219 Graphs are nearly impossible to read bc they are too small. It seems like not all post hoc comparisons are labeled - however, it might just be too blurry.

-We apologize that our graphs were given at too low of a resolution. We ensured that images of 600 DPI were uploaded.

  1. Line 219 Consider reorganizing the graph to be hours post drinking with the ZTs as my understanding is your brain samples are only taken starting at ZT 15 (according to Line 146).

-We feel presenting data in terms of zeitgeber time is of more use to the fields of alcohol and circadian research.

  1. Page: 6 - Line 223 Figure captions should not repeat reporting of the statistics, ie the F and p values. Simply describe the graph and state the outcome. At the end define the * meaning(s) for all graphs; this should be consistent throughout, ie p<.05 is always *, p<.01 is always **, etc. This lengthy of a figure caption is difficult to read. Also, please bold the graph letter, ie (A).

-We have presented data in this same fashion in several publications and feel the transparency of our current figure captions to be important for the reader.

  1. Line 243 Please list ALL exact, p values. You can keep * and ns, but after the listed p-value, ie p=.21, ns or p=.01, **

-The current style of presenting p-values is well published in our lab and we feel this is a more readable approach.

  1. Page: 7 - Line 254 List type of analysis including factors and levels

-Please see response to item 24 above.

  1. Line 257 You should include data for water intake as well as it could be affected by PF-670462.

-There was no effect of PF-670462 on water drinking, therefore we decided that only presenting the alcohol data was simpler and made comparisons across behavioral pharmacology studies (experiments 2 and 3) easier for the reader. 

  1. Line 267 Again, do not repeat methods or design; just state results and include statement of analysis type with factors and levels.

-Items This comment is very similar to earlier concerns. In particular, please see our response to items 23.

  1. Page: 8 - Line 286 Statistical sig not indicated for interaction in graph.

-This detail has been provided.

  1. Page: 9 - Line 302 Again, do not repeat methods or design; just state results and include statement of analysis type with factors and levels.

-Please see response to items 23 and 34.

  1. Line 315 If you included the previous day 4 times, you would not see a drug effect on weeks 7 & 8 as those mice also show decreased drinking compared to controls despite them being treated the same up to week 7. By not stating the factors and levels it is unclear that you are not comparing all time points which is not appropriate.

-This concern is reflected in the above items (namely 24 and 30) and we feel we have adequately addressed this concern in response to them.

  1. Line 317 p value typo - should be p< .01 (after rounding).

-This correction has been made

  1. Line 329 Again, do not repeat methods or design; just state results and include statement of analysis type with factors and levels.

-Please see response to items 23, 34 and 36.

  1. Line 339 It is not appropriate to include a post hoc test when there is not an interaction.

-We appreciate that this concern was brough to our attention. In response, care and attention was provided to ensure that post-hoc testing was only performed on

  1. Page: 11 - Line 376 Consider you are measuring gene expression and not protein changes after treatment and this timeline may be different. You could also consider activity based changes in proteins (ie phosphorylation of your target sites).

-This is a valid concern and this is now addressed as a limitation in the discussion section.

  1. Line 386 DID also has a two-bottle method and you are using a rodent line that has increased intake. Your phrasing should indicate a desire to maximize consumption rather than suggesting the other models do not reach physiological levels of intake (which they often do).

-In response, we have included references which better support this sentence.  

  1. Line 391 Is this a dependence model? Note the contrast to binge.

-The authors of this work do not explicitly frame their paradigm as a measure of dependence, therefore we chose to similarly reflect that this design was a mode of “chronic ethanol consumption”:  

  1. Line 398 This transition is clunky. Consider starting with the current findings instead of previously published work.

-A less clunky transition has been provided.

  1. Line 403 Again, when discussing specific findings, please start with the findings from the current studies and provide supporting or contrasting citations as a follow up. It is fine to establish a topic with widely accepted statement supported by literature (like in the next paragraph) but it's awkward to start a discussion paragraph relaying specific findings from a different paper not from the current lab.

-We agree with the reviewer that this transition is somewhat clunky. We have added a transition which highlights the contrast of this work in comparison to our own.

Reviewer 3 Report

This is a high quality study that documents changes in circadian clock gene expression in three key areas of the brain in genetically defined mice bred as a model for binge drinking of alcohol. One drug that inhibits CK altered circadian clock gene expression but not drinking behavior, and a second drug that inhibits PDE altered drinking behavior but not clock gene expression.  This study is of particular interest because it uses the HDID mouse model for binge drinking, documents extensive clock gene expression effects of alcohol, and provides an interesting basis for further study of possible mechanisms linking clock gene expression to alcohol use disorders.  Minor revisions are suggested to reduce redundancy in the results section and to consider points that may have been  overlooked in the discussion.

1. The main point the authors might address more clearly in the abstract and discussion is that their drug treatment results suggest that the changes in gene expression they observe are not related to binge-drinking of alcohol. While it is fine to note  and discuss differences between the results of this study and those of previous studies that seem inconsistent, the authors seem to have overlooked the fact that their own experiment suggests that the gene expression changes they observed are not related to binge drinking: the foundation of the study is changes in gene expression in response to binge-drinking. The obvious question this raises is whether the gene expression changes identified mediate the behavior being modeled (binge drinking). The changes in circadian gene expression could be either causally related to the binge drinking phenotype or incidental. The two drug treatments provide an independent test of the relationship between alcohol consumption and circadian gene expression. Since one drug reduced alcohol consumption but didn't alter gene expression, and the other drug altered gene expression but didn't change drinking behavior, the author's drug treatments failed to demonstrate any causal connection between changes in circadian gene expression and alcohol consumption.  This is not a problem in itself: the CK inhibitor effects on gene expression only affected on of the three brain regions, for example; similarly, the PDE inhibitor altered drinking but not gene expression- it may act through a different mechanism, or downstream from circadian clock effects. These negative findings (with respect to connecting the clock gene expression changes with alcohol consumption behavior) should be noted as such, but do not detract from the overall value of this model for exploring these mechanisms further, or from the documentation of clock gene expression changes in amplitude caused by the alcohol treatment. 

2) A second point the authors might consider for discussion is more subtle. Although the author's rationale for using the HDID mice as a model here is that they are genetically bred for binge-like drinking, this may also be a reason not to use them. If the behavior is driven by changes in expression of clock genes in response to drinking, the many generations of artificial selection may have bred those changes into these mice so that they occur even without drinking (e.g "genetic assimilation").  The changes induced at this point may be relatively minor compared to what may have already been selected for and constitutively expressed, even in the mice drinking water.  It may be the case, for example, that the binge drinking is mediated through changes in circadian phase, period and amplitude, and phase and period changes may have been genetically maxed out, but there is still some capacity for response for amplitude since that is less dependent on circadian clock function. The only way to know what changes in circadian gene expression may mediate binge drinking in the HDID selected line is to compare it to an unselected control line or a line selected for decreased alcohol consumption, notwithstanding confounding effects of genetic drift in artificial selection experiments that randomly fix genetic variation across selected lines. In other words, the author's rationale for using these mice this way assumes depends on the genetic predisposition to binge drink being mediated through gene regulation by alcohol, but it may be that baseline levels of gene expression have already been altered by artificial selection in these mice to a point where further regulation in response to alcohol consumption is limited by a ceiling effect imposed by selection for high constitutive levels of expression relevant to alcohol consumption. This is also not a weakness of the study- just something that could, logically, be considered and explored further in future studies (e.g. by comparing HDID mice to unselected controls or low-consumption lines). 

3. The only other non-trivial point is that the statistical results are presented twice in the Results section: once in the figure legend and again in tables. One or the other is sufficient. If tables are retained the tables could be more concise (e.g. the F value and df are the same for each row, so they can go in the main-effect column instead of every cell in the row). 

4. Abstract, line 11: change "there minimal" to "there is minimal"?

5. Intro, line 59: change "importance of molecular clock" to "importance of this molecular clock"?

6. Intro, line 60: it's not clear here what the authors mean by "compensatory effects": compensatory to what, and how? This becomes clearer in the discussion (lines 413-414) and is an interesting point. 

7. Intro, lines 93-94: how about "in the rescue or amelioration of..."?

8. Intro, line 95: I understand "predictive validity", but maybe what the authors mean by "construct" and "face" could be explicated a bit?

9. Methods, lines 138-139: it would be clearer to state, explicitly, that "n" mice were given water and "n" mice alcohol for the duration of the experiment. Saying that "...a single bottle of 20% ethanol or water was provided...." could mean that a given mouse received water one week and ethanol another.

10. Discussion, line 364: change "Clockdelta19 (which have a ....." to "Clockdelta19 mice (with a ...."?

one drug decreased drinking but didn't affect gene expression, and the

Author Response

IJMS Reviewer comments

Reviewer 3:

This is a high quality study that documents changes in circadian clock gene expression in three key areas of the brain in genetically defined mice bred as a model for binge drinking of alcohol. One drug that inhibits CK altered circadian clock gene expression but not drinking behavior, and a second drug that inhibits PDE altered drinking behavior but not clock gene expression.  This study is of particular interest because it uses the HDID mouse model for binge drinking, documents extensive clock gene expression effects of alcohol, and provides an interesting basis for further study of possible mechanisms linking clock gene expression to alcohol use disorders.  Minor revisions are suggested to reduce redundancy in the results section and to consider points that may have been overlooked in the discussion.

  1. The main point the authors might address more clearly in the abstract and discussion is that their drug treatment results suggest that the changes in gene expression they observe are not related to binge-drinking of alcohol. While it is fine to note  and discuss differences between the results of this study and those of previous studies that seem inconsistent, the authors seem to have overlooked the fact that their own experiment suggests that the gene expression changes they observed are not related to binge drinking: the foundation of the study is changes in gene expression in response to binge-drinking. The obvious question this raises is whether the gene expression changes identified mediate the behavior being modeled (binge drinking). The changes in circadian gene expression could be either causally related to the binge drinking phenotype or incidental. The two drug treatments provide an independent test of the relationship between alcohol consumption and circadian gene expression. Since one drug reduced alcohol consumption but didn't alter gene expression, and the other drug altered gene expression but didn't change drinking behavior, the author's drug treatments failed to demonstrate any causal connection between changes in circadian gene expression and alcohol consumption.  This is not a problem in itself: the CK inhibitor effects on gene expression only affected on of the three brain regions, for example; similarly, the PDE inhibitor altered drinking but not gene expression- it may act through a different mechanism, or downstream from circadian clock effects. These negative findings (with respect to connecting the clock gene expression changes with alcohol consumption behavior) should be noted as such, but do not detract from the overall value of this model for exploring these mechanisms further, or from the documentation of clock gene expression changes in amplitude caused by the alcohol treatment. 

-We greatly appreciate that this distinction was clarified by the reviewer. In response, the following limitations have been highlighted in the discussion section:

“The effect of PF-670462 on behavioural measures of circadian phase and relapse-like drinking is in direct contrast to our own findings, whereby CK1e/d inhibition had no effect on ethanol intake. Future work should address circadian gene expression in models of harmful drinking across multiple timepoints (similar to experiment 1) and in several relevant drinking paradigms, species, and strains.”

“Taken together, the dissonance between the effects of PF-670462, which changed circadian expression, but not ethanol intake, in contrast to apremilast, which reduced drinking despite no effects on circadian genes, highlights an important limitation in our understanding between the role of circadian gene expression and binge-like ethanol drinking.”

  1. A second point the authors might consider for discussion is more subtle. Although the author's rationale for using the HDID mice as a model here is that they are genetically bred for binge-like drinking, this may also be a reason not to use them. If the behavior is driven by changes in expression of clock genes in response to drinking, the many generations of artificial selection may have bred those changes into these mice so that they occur even without drinking (e.g "genetic assimilation").  The changes induced at this point may be relatively minor compared to what may have already been selected for and constitutively expressed, even in the mice drinking water.  It may be the case, for example, that the binge drinking is mediated through changes in circadian phase, period and amplitude, and phase and period changes may have been genetically maxed out, but there is still some capacity for response for amplitude since that is less dependent on circadian clock function. The only way to know what changes in circadian gene expression may mediate binge drinking in the HDID selected line is to compare it to an unselected control line or a line selected for decreased alcohol consumption, notwithstanding confounding effects of genetic drift in artificial selection experiments that randomly fix genetic variation across selected lines. In other words, the author's rationale for using these mice this way assumes depends on the genetic predisposition to binge drink being mediated through gene regulation by alcohol, but it may be that baseline levels of gene expression have already been altered by artificial selection in these mice to a point where further regulation in response to alcohol consumption is limited by a ceiling effect imposed by selection for high constitutive levels of expression relevant to alcohol consumption. This is also not a weakness of the study- just something that could, logically, be considered and explored further in future studies (e.g. by comparing HDID mice to unselected controls or low-consumption lines). 

-The authors agree with this comment and feel this is an important limitation to address. In response, the following sentence was added to the final paragraph of the discussion:

“Moreover, this suggests that HDID-1 and iHDID-1 may have important baseline differences in brain-wide circadian gene expression patterns and may respond differently to harmful alcohol intake compared to other strains. Future work looking at the interface of stress and alcohol should also consider the importance of testing multiple timepoints (such as the 8 timepoints tested in the present study) and the use of unique genetic models of ethanol-related behaviors with known differences in circadian phenotypes”

  1. The only other non-trivial point is that the statistical results are presented twice in the Results section: once in the figure legend and again in tables. One or the other is sufficient. If tables are retained the tables could be more concise (e.g. the F value and df are the same for each row, so they can go in the main-effect column instead of every cell in the row). 

-This concern reflects several that were brought up by reviewer 1 and we feel that we have adequately responded to this comment in our above responses. 

  1. Abstract, line 11: change "there minimal" to "there is minimal"?

            -This change has been made

  1. Intro, line 59: change "importance of molecular clock" to "importance of this molecular clock"?

            -This change has been made

  1. Intro, line 60: it's not clear here what the authors mean by "compensatory effects": compensatory to what, and how? This becomes clearer in the discussion (lines 413-414) and is an interesting point. 

-We agree with the reviewer that the use of “compensatory effects” in line 60 is vague. In response, this sentence has been changed as follows:

“It is difficult to know whether the heightened ethanol drinking displayed by mutant mice results from circadian genes directly or from unknown compensatory mechanisms, such as downstream effects on reward-related processing.”  

  1. Intro, lines 93-94: how about "in the rescue or amelioration of..."?

-This change has been made.

  1. Intro, line 95: I understand "predictive validity", but maybe what the authors mean by "construct" and "face" could be explicated a bit?

-A brief description of each type of validity have been added to this section as follows:

“ HDID mice are a valuable genetic animal model with construct (maintains descriptive power for human AUD, i.e., accurately captures drinking to pharmacological intoxicating), face (resembles human binge drinking), and predictive (used to accurately predict human binge drinking outcomes) validity [26, 27].”

  1. Methods, lines 138-139: it would be clearer to state, explicitly, that "n" mice were given water and "n" mice alcohol for the duration of the experiment. Saying that "...a single bottle of 20% ethanol or water was provided...." could mean that a given mouse received water one week and ethanol another.

-Per this suggestion, , we have explicitly stated the “n” per condition.

  1. Discussion, line 364: change "Clockdelta19 (which have a ....." to "Clockdelta19 mice (with a ...."? one drug decreased drinking but didn't affect gene expression, and the

            -We were unable to decipher what this comment sought to address.  
